# Incidence and Risk Factors of Venous Thromboembolic Events in Patients with ANCA-Glomerulonephritis: A Cohort Study from the Maine-Anjou Registry

**DOI:** 10.3390/jcm9103177

**Published:** 2020-09-30

**Authors:** Nicolas Henry, Benoit Brilland, Samuel Wacrenier, Assia Djema, Anne Sophie Garnier, Renaud Gansey, Jean-Philippe Coindre, Virginie Besson, Agnès Duveau, Jean-François Subra, Maud Cousin, Giorgina Barbara Piccoli, Jean-François Augusto

**Affiliations:** 1Service de Néphrologie-Dialyse-Transplantation, Université d’Angers, CHU Angers, 4 rue Larrey, 49033 Angers CEDEX 09, France; nicolas.henry@chu-angers.fr (N.H.); benoit.brilland@chu-angers.fr (B.B.); samuel.wacrenier@chu-angers.fr (S.W.); annesophie.garnier@chu-angers.fr (A.S.G.); vibesson@chu-angers.fr (V.B.); agnes.duveau@chu-angers.fr (A.D.); jfsubra@chu-angers.fr (J.-F.S.); macousin@chu-angers.fr (M.C.); 2CHU d’Angers, Université d’Angers, INSERM U1232, CRCINA, 49000 Angers, France; 3Service de Néphrologie-Dialyse, CH du Mans, 72000 Le Mans, France; jpcoindre@ch-lemans.fr (J.-P.C.); gbpiccoli@yahoo.it (G.B.P.); 4Service de Néphrologie-Dialyse, CH de Cholet, 49300 Cholet, France; assia.djema@ch-cholet.fr; 5Service de Néphrologie-Dialyse, CH de Laval, 53000 Laval, France; renaud.gansey@chlaval.fr

**Keywords:** venous thromboembolism, ANCA glomerulonephritis, statins, risk factor

## Abstract

(1) Introduction: The incidence of venous thromboembolisms (VTE) has not been extensively analyzed in patients with antineutrophil cytoplasmic antibody (ANCA)-glomerulonephritis (ANCA-GN). Thus, the aim of the present study was to assess the frequency and the risk factors of VTE in patients with ANCA-GN. (2) Methods: Patients from the Maine-Anjou ANCA-associated vasculitis (AAV) registry with a biopsy showing pauci-immune glomerulonephritis were included. VTE events, site, and interval from AAV diagnosis were analyzed. (3) Results: 133 patients fulfilled the inclusion criteria of the study and were analyzed. VTE episodes were diagnosed in 23/133 (17.3%) patients at a median delay of 3 months from ANCA-GN diagnosis. Patients with VTE had lower serum albumin (*p* = 0.040), were less frequently on statin therapy (*p* = 0.009) and had less frequently proteinase-3 (PR3)-ANCAs (*p* = 0.078). Univariate analysis identified higher age (*p* = 0.022), lower serum albumin (*p* = 0.030), lack of statin therapy (*p* = 0.009), and rituximab treatment (*p* = 0.018) as significant risk factors of VTE. In multivariate analysis, only lack of statin therapy (HR 4.873; *p* = 0.042) was significantly associated with VTE. (4) Conclusion: Patients with ANCA-GN are at high risk of VTE, especially within the first months following AAV diagnosis. Our results suggest that statin therapy is associated with a lower risk of VTE in ANCA-GN patients.

## 1. Introduction

Venous thromboembolism (VTE) is a classical complication of chronic inflammatory diseases [1], reported to be increased in many auto-immune and rheumatic diseases, including ANCA-associated vasculitis (AAV) [2,3,4,5,6,7,8,9]. The first studies specifically addressed in AAV were a case series of pediatric patients that developed deep vein thrombosis (DVT) shortly after disease diagnosis [10] and a post-hoc analysis of the Wegener’s Granulomatosis Etanercept Trial (WGET) [2,11]. In this prospective trial, which was limited to patients with granulomatosis with polyangiitis (GPA), about 10% of patients developed VTE after a median time of 2.1 months from disease onset [11]. The increased risk of VTE in AAV patients was confirmed in further retrospective studies [4,5]. Stassen et al. reported an incidence of 12% of VTE in a cohort of 193 patients with micropolyangitis (MPA) or GPA, followed for 6.1 years [4]. A similar incidence was observed in a recent analysis performed in 417 patients that were included in randomized controlled trials conducted by the European Vasculitis Society [5]. In this analysis, patients with severe renal disease had a higher rate of VTE. In the multivariate analysis, increased C-reactive protein level and increased creatinine cutaneous and gastrointestinal involvement at diagnosis were independently associated with VTE. However, the analysis of VTE risk factors was limited given the heterogeneity of patients pooled from four different studies, and by unavailability of detailed data at disease onset. Kronbichler et al. analyzed VTE frequency and risk factors in patients included in the RAVE study which aimed at evaluating rituximab as a remission induction regimen [9,12]. In these patients without severe kidney involvement, pulmonary hemorrhage, and positive PR3-ANCA, heart involvement and microscopic hematuria were found to be independently associated with VTE development. More recently, Isaacs et al. found PR3-ANCA, hypoalbuminemia, and BMI to be independent risk factors of VTE in a retrospective analysis in 162 AAV patients [13].

Kidney involvement in AAV appears as a major prognostic factor [14] and impaired kidney function has been reported as a risk factor of VTE [5]. Finally, no study has specifically addressed VTE risk factors in ANCA-glomerulonephritis (ANCA-GN) patients and whether factors identified in past studies apply to ANCA-GN patients. Moreover, whether ANCA-GN patients experiencing VTE have a different prognosis remains to be studied. 

The AAV Maine-Anjou registry includes all patients diagnosed with ANCA-GN from the Maine Anjou Region in France, encompassing one university and three regional hospitals [15]. The registry gathers very detailed clinical and biological data of patients with ANCA-GN since 2000, as well as treatment management and outcomes. 

Thus, in the present study, we used the AAV Maine-Anjou registry to study VTE episodes in patients with biopsy confirmed ANCA-GN. The primary objective was to analyze the incidence of VTE in ANCA-GN patients and the secondary objectives were to identify risk factors of VTE and to study the outcome of patients according to VTE occurrence.

## 2. Material and Methods

### 2.1. Maine-Anjou Registry

The Maine-Anjou registry is a retrospective-prospective registry which was created on 1 January 2018. The database includes adult patients with ANCA-GN diagnosed since 1 January 2000 in Nephrology departments of 4 hospitals (Angers University Hospital and Regional Hospital of Le Mans, Cholet and Laval). Patients are included in the registry if aged over 18-years-old, if they fulfill Chapel Hill Consensus Conference criteria for AAV [16], and have presumed or confirmed renal involvement of vasculitis. Data concerning characteristics, presentation (clinical and biological data), treatment, and outcomes are collected in the registry. Data were collected retrospectively before 1 January 2018 and prospectively after 1 January 2018. On the 1 January 2019, the registry had data from 165 patients.

The registry has been declared and has been authorized by the “Commission National Informatique et Liberté” (CNIL, agreement number 2018-MR03-02). In accordance with French law, participants gave their non-opposition to be included in the registry and for the use of their data in an anonymous form.

### 2.2. Inclusion Criteria in the Study and Data Collection

Patients from the registry were eligible for inclusion in the study if they had pauci-immune glomerulonephritis confirmed at kidney biopsy and a follow-up of at least 6 months. Patients were followed from ANCA-GN diagnosis until end of follow-up or death. Relapsing patients had ANCA-GN at relapse and were followed from relapse to the end of follow-up. VTE, their sites, and their timing according to ANCA-GN onset were collected, as well as risk factors of VTE, antiplatelet or/and anti-thrombotic ongoing treatments at ANCA-GN diagnosis. Thrombotic episodes of the arterio-venous fistulae for hemodialysis were excluded.

Data were retrieved from the registry: Age, gender, weight, height, and comorbidities; the nature and type of injuries of the affected organs at AAV presentation were listed. The AAV activity was determined using the Birmingham Vasculitis Activity Score (BVAS) 2003 [17]. Biological parameters at diagnosis were also retrieved, as well as medications at AAV diagnosis. The glomerular filtration rate was calculated using the 4-variable Modification of Diet in Renal Disease (MDRD) study equation [18]. 

Kidney biopsy results were also retrieved when available. All kidney biopsies from the four centers are analyzed centrally in the department of Pathology of the University Hospital of Angers by two nephropathologists. Only biopsies showing more than 7 glomeruli were considered in the present study. The analysis of kidney biopsies is routinely reported in a standardized pathological report allowing classification according to the Berden histopathological classification of ANCA-GN [14].

The study protocol complied with the Ethics Committee of the Angers University Hospital (n 2020/84).

### 2.3. Treatments and Definitions

The therapeutic regimens used to induce and maintain AAV remission was left at the clinician discretion. AAV subtype (GPA and MPA) was classified according to the European Medicines Agency (EMA) vasculitis classification algorithm [19]. Renal disease characterization was based on kidney biopsy and clinical data (active urinary sediment, proteinuria, and impaired renal function). Renal death was defined as the need for renal replacement therapy (RRT) for more than 3 months. VTE episodes were considered if confirmed by ultrasounds or CT scan. The prognosis associated with the occurrence of VTE following the first 3 months since ANCA-GN diagnosis (early VTE) was also evaluated. 

### 2.4. Statistical Analysis

Continuous variables are presented as the mean ±SD or median and interquartile range (IQR) when applicable. Categorical variables are presented as the absolute value and percentage. Differences between groups were analyzed using the χ^2^ test (or Fisher exact test when applicable) for categorical variables and the Mann–Whitney U test for continuous variables. The Kaplan–Meyer method was used to analyze VTE-free survival. Cox proportional hazards regression analysis was performed to examine factors associated with VTE and are reported as hazard ratio (HR) with 95% CIs. Multivariate Cox regression analysis included all parameters that were correlated with VTE occurrence in the univariate analysis (*p* < 0.05). All the statistical tests were performed to the two-sided 0.05 level of significance. Statistical analysis was performed using SPSS software^®^ 23.0 (IBM, Armonk, NY, USA) for Mackintosh and Graphpad Prism^®^ 7 (San Diego, CA, USA).

## 3. Results

### 3.1. Characteristics of the Population

Among 165 patients included in the Maine-Anjou AAV registry, 145 underwent a kidney biopsy at AAV onset or at relapse. Among these 145 patients, 141 had a kidney biopsy showing pauci-immune glomerulonephritis. Eight patients were excluded because of missing data and/or unavailable follow-up. Thus, 133 patients were included in the present study (Figure 1). The mean age at presentation was 65.1 ± 14.1 years with a predominance of males (63.2%), 45 patients had GPA and 88 MPA. MPO-ANCAs were detected in 86 (64.7%) patients and PR3-ANCAs in 39 (29.3%) patients. Eight patients were MPO/PR3 ANCA negatives.

BVAS was 17.2 ± 6.1 at disease onset or relapse. Median serum creatinine and eGFR at presentation were 279.0 μmol/L and 20.7 mL/min/1.73 m^2^, respectively.

Cyclophosphamide was predominantly used as remission induction treatment, all patients received steroids and 38 (28.6%) were treated with plasma exchange. The median follow-up of the cohort was 40.5 months. 

During follow-up, 35 (26.3%) patients developed end-stage renal disease (ESRD) and 26 (19.5%) died. Baseline characteristics of patients at the time of kidney biopsy are presented in Table 1.

### 3.2. Venous Thromboembolic Events

We first analyzed the frequency and sites of VTE. Eight patients (6%) had history of VTE before ANCA-GN diagnosis. Twenty-five VTE occurred in 23 patients (17.3%) during the follow-up. The median delay to first VTE was 3.03 months. VTE were predominantly DVT with leg localization, while pulmonary embolism occurred in 7 patients (5.2%). In 4 cases, DVT developed as a complication of venous catheter which was required for hemodialysis. The estimated one-year incidence of VTE was 12.4%. These data are reported in Table 2. Figure 2 reports the survival free of VTE in the study population.

Anticardiolipid antibodies, anti-beta2 GP1 antibodies, and lupus anticoagulants were searched in 64 (48.1%), 55 (41.3%), and 41 (30.8%) patients, respectively. In those that had at least one test (*n* = 67), abnormality of at least one test was detected in 8 patients (11.9%). These data are reported in Appendix A.

### 3.3. Comparison of Patients According to VTE Occurrence

Patients that experienced at least one VTE had lower serum albumin concentration (*p* = 0.040) and were less frequently on statin therapy at AAV diagnosis (*p* = 0.009). Age was not significantly different (*p* = 0.081) between groups, and no significant difference was observed according to gender, AAV phenotype, and ANCA subtype. At ANCA-GN diagnosis, AAV activity based on BVAS and C-reactive protein level was comparable between groups and no difference in organ involvement was observed. Remission induction treatment was also not significantly different and plasma exchange use was comparable between groups. These data are reported in Table 3.

### 3.4. Risk Factor of VTE Occurrence

In the univariate analysis, age, serum albumin concentration, statin therapy at ANCA-GN diagnosis, and rituximab induction were significantly associated with the risk of developing VTE. In the multivariate analysis, only lack of statin therapy at AAV diagnosis was significantly associated with VTE occurrence (HR 4.73, *p* = 0.042), while the association with serum albumin was borderline (HR 2.26, *p* = 0.057). Age and rituximab treatment were no longer associated with VTE after adjustment. These results are presented in Table 4.

Apart from being more frequently on antiplatelet agents, patients on statin therapy did not significantly differ at ANCA-GN onset as compared to those without statins (Appendix A). Moreover, the rate of statin treatment was not significantly different between centers (Appendix A).

### 3.5. Prognosis of AAV Patients with VTE

In a last step, we analyzed the prognosis associated with VTE occurrence within the first 3 months (early VTE) following ANCA-GN diagnosis. We observed that patients with early VTE (*n* = 12) developed more frequently ESRD (66.6% (*n* = 8) versus 22.3% (*n* = 27), *p* = 0.001). However, no relationship between early VTE and death (VTE group 25.0% (*n* = 3) versus 19.0% (*n* = 23), *p* = 0.628) or AAV relapse (VTE group 33.3% (*n* = 4) versus 29.7%, *p* = 1.000) was observed. Given that kidney histological involvement is associated with renal outcome [14], we searched for an association between renal histopathology and VTE occurrence. However, we did not observe any difference in histological injury between patients with and without VTE within the first 3 months following ANCA-GN diagnosis, as assessed using Berden classification (Figure 3).

## 4. Discussion

In the present study, we confirm a high incidence of VTE in ANCA-GN patients. Indeed, VTE occurred in 17.3% of patients at the end of follow-up, representing an estimated one-year incidence of 12.4%. While the univariate analysis identified age, low serum albumin at ANCA-GN onset, lack of statin therapy, and rituximab as significant risk factors of VTE; in the multivariate analysis, only statin therapy was significantly associated with a lower risk of VTE. To the best of our knowledge, this observation suggesting antithrombotic action of statins has not yet been reported in ANCA-GN patients and may open new therapeutic perspectives.

The antithrombotic action of statin therapy has been observed in human prospective randomized clinical trials [20,21]. A recent meta-analysis comprising more than three million participants from cohort studies and randomized control trials showed a significant risk reduction of DVT with a RR of 0.75 in patients on statin therapy as compared to patients on placebo [22]. Interestingly, in this meta-analysis, a greater reduction risk of VTE was observed in the subgroup of patients with higher VTE risk (RR 0.46). Potential protective actions of statins may be explained by several ways. Indeed, several in vitro and in vivo animal and human studies have demonstrated an impact of statins on blood coagulation [23,24]. In addition to their anti-inflammatory action, statins have been shown to favor an anticoagulant state by reducing tissue factor expression, enhancing the protein C activity by increasing thrombomodulin expression on endothelial cells [25], and by decreasing plasminogen activator inhibitor 1 and increasing tissue plasminogen activator [26]. In the present study, lack of statin therapy conferred a 4.7-fold risk of developing VTE in our cohort. Thus, given that anti-thrombotic properties of statins may be stronger in patients at high VTE risk, we suggest that their anti-thrombotic action may be greatly enhanced in ANCA-GN patients, which could account for our results. Interestingly, patients under statin therapy at AAV diagnosis were more frequently under antiplatelet agents as compared to patients without statins. However, no difference was present as regards to hypertension or diabetes, nor to cholesterol concentration at ANCA-GN diagnosis between patients with and without statins. Moreover, the rate of statin therapy was not significantly different between centers suggesting the lack of a center effect.

In the two major previous studies that analyzed VTE in AAV patients, the incidence of VTE was 9.8% and 12%, thus lower than the incidence observed in the present study [2,4]. In the study from Kronbichler et al., there was a trend towards a higher incidence of VTE in the MEPEX trial that included patients with severe renal disease [5], which is in accordance with the higher incidence of VTE in our study. The differences in VTE incidences may also be explained by heterogeneity in disease presentation, severity, and stages of disease (flare versus remission) of patients included in past studies [2,4].

We observed that most VTE occurred soon after AAV diagnosis when patients had active vasculitis. The median delay to VTE was about 3 months after ANCA-GN diagnosis, which corresponds to the remission-induction therapeutic phase. Given that no systematic VTE screening was performed in our study, it is probable that the true delay between AAV diagnosis and VTE occurrence may be shorter. Thus, most VTE occurred in a period when several risk factors of VTE are present, such as hospitalization, immobilization related to disease severity, and the need for central venous access for hemodialysis or plasmapheresis.

We were not able to study the role for endogenous hypercoagulable factors in our study. Only one third of patients were screened for APL and/or LA, not allowing to investigate relationship with VTE. We observed a low prevalence of APL and lupus anticoagulant in screened patients. 

Interestingly, we observed that patients that experienced VTE within the first 3 months following AAV diagnosis developed more frequently ESRD. The kidney involvement of these patients as assessed by kidney biopsy was not significantly different as compared to other patients. We also did not observe any significant difference in treatment regimens that could explain this observation. Unfortunately, the low number of patients with this condition does not allow us to analyze causative factors extensively.

Interestingly, serum albumin appeared as a potential risk factor for VTE in ANCA-GN patients in our study. Indeed, we observed an association between serum albumin at baseline with an HR of 2.42 for each 10 g/L albumin decrease in the univariate analysis. The relationship between hypoalbuminemia and VTE is well documented in patients with nephrotic syndrome, where hypoalbuminemia is mainly the consequence high albuminuria [27,28], and has also been evidenced as an independent risk factor of VTE in a recent study conducted in AAV patients with and without renal involvement [13]. In our study, proteinuria was not different between patients with and without VTE, suggesting that mechanisms other than proteinuria may be involved to explain more profound hypoalbuminemia in VTE patients. Finally, serum albumin was no longer associated with VTE in the multivariate analysis, maybe because our study was underpowered. Further studies should focus on this aspect to better understand the factor favoring venous thrombosis in ANCA-GN patients.

In contrast to previous studies [5], we did not observe any association between VTE and AAV organ involvement, C-reactive protein level or BVAS at presentation. This may be explained by our design that only included patients with ANCA-GN with finally poor representation of patients with other organ involvement. The lack of association with inflammation may also be related to the fact that patients of our study had high disease activity at presentation mainly driven by their renal involvement (median BVAS 17.2), limiting any comparison between patients with severe and milder disease activity. In a recent post-hoc analysis of the RAVE trial [12], Kronbichler et al. showed that heart involvement, pulmonary hemorrhage, PR3-ANCA, and microscopic hematuria were independently associated with VTE development. However, it is important to underline that patients with serum creatinine above 354 µmol/L were excluded from RAVE study [12] and that kidney involvement of RAVE study patients was mild with a mean eGFR above 50 at baseline as compared to 20 mL/min/1.73 m^2^ in our study. Finally, we were not able to identify “classical” risk factors found in previous studies, but found association with novel factors not yet or poorly reported. This may be related to the fact that patients in our cohort had a very detailed phenotype allowing us to analyze a large number of potential risk factors. Another explanation may be that that classical risk factors of VTE could be overcome in the context of ANCA-GN.

Our study has several limitations starting with its observational design. We did not perform systematic radiological screening of DVT, thus some VTE may have been missed. Moreover, we did not perform systematic screening of blood clotting and of APL antibodies and the use of thromboprophylaxis at ANCA-GN diagnosis was not collected. However, despite these limitations, this study is the first to analyze specifically the incidence and risk factors of VTE in a well characterized population of patients with ANCA-GN.

In conclusion, in line with previous studies, we show a high incidence of VTE in ANCA-GN patients, predominantly within the first months following ANCA-GN diagnosis. The present study identifies statin therapy as an independent risk factors of VTE in ANCA-GN patients, thus suggesting a protective role of statins and opening interesting research and therapeutic perspectives. Furthermore, these results give some suggestions for future research that should focus on the effect of albumin levels on VTE risk and on the impact of experiencing VTE on renal outcome. Thus, these results may help identifying ANCA-GN patients with higher VTE risk and defining specific preventive strategies.

## Figures and Tables

**Figure 1 jcm-09-03177-f001:**
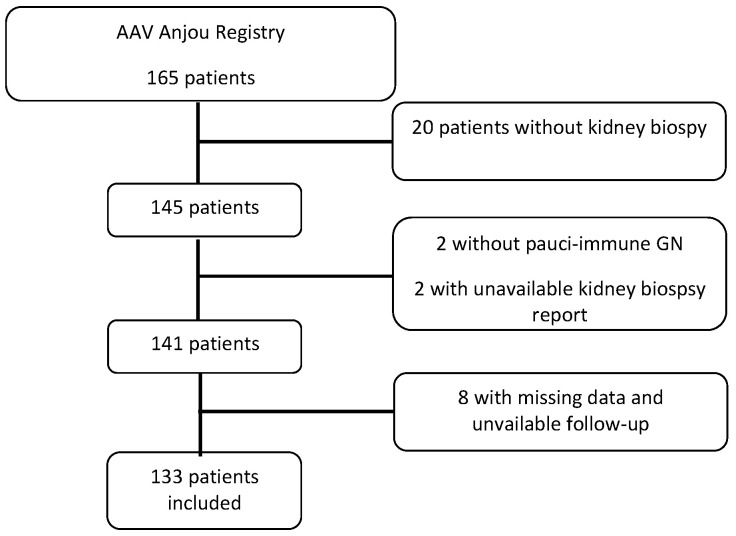
Flowchart of the study.

**Figure 2 jcm-09-03177-f002:**
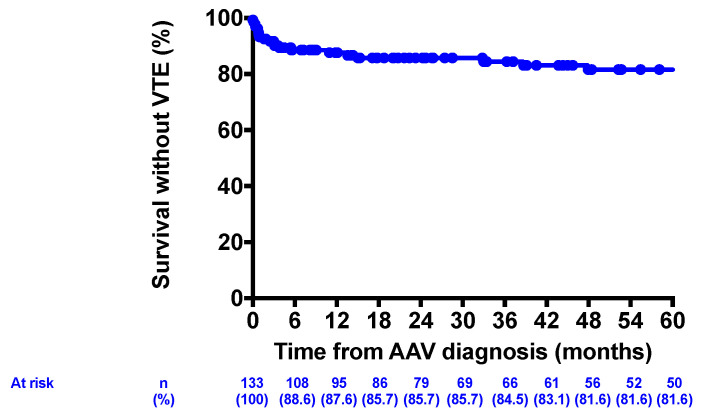
Survival free of thromboembolic events.

**Figure 3 jcm-09-03177-f003:**
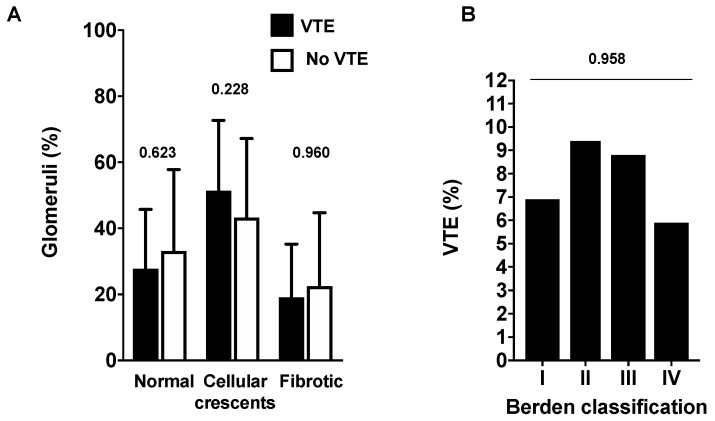
Kidney biopsy histology according to early venous thromboembolism (VTE) occurrence. (**A**) percentage of normal, crescentic, or sclerotic glomeruli in patients with and without early VTE. (**B**) percentage of patients with VTE according to Berden’s classification. I, focal; II, crescentic; III, mixed; and IV, sclerotic form.

**Table 1 jcm-09-03177-t001:** Baseline characteristics of the population and outcome.

	All, *n* = 133
**Baseline characteristics**	
Gender, M/F	84/49
Age, years	65.1 ± 14.1
BMI, kg/m^2^	24.6 ± 4.5
BMI > 30 kg/m^2^, *n* (%)	14 (10.5)
Heart disease, *n* (%)	15 (11.3)
Atrial fibrillation, *n* (%)	7 (5.3)
Past history of cancer, *n* (%)	19 (14.3)
**ANCA-associated vasculitis characteristics**	
**Clinical diagnosis, *n* (%)**	
Newly diagnosed	122 (91.7)
GPA/MPA	45 (33.8)/88 (66.2)
**ANCA subtype, *n* (%)**	
PR3 ANCA	39 (29.3)
MPO ANCA	86 (64.7)
ANCA negative	8 (6.0)
**BVAS at kidney biopsy**	17.2 ± 6.1
**Organ involvement at diagnosis**	
Cutaneous signs, *n* (%)	25 (18.8)
Ear, nose, throat, *n* (%)	46 (34.6)
Heart, *n* (%)	8 (6.0)
Digestive, *n* (%)	7 (5.3)
Lung, *n* (%)	48 (36.1)
Neurological, *n* (%)	19 (14.3)
Renal (at kidney biopsy)	
Serum creatinine, µmol/L (ICQ)	279.0 (133–419)
eGFR, mL/min/1.73 m^2^ (ICQ)	20.7 (11–43)
Proteinuria, g/g (ICQ)	1.67 (0.8–2.9)
Proteinuria > 3g/g, *n* (%)	31 (23.3)
Need for renal replacement therapy, *n* (%)	30 (22.6)
**Induction remission regimen, *n* (%)**	
Cyclophosphamide	116 (87.2)
Rituximab	7 (5.3)
Other	10 (7.5)
Plasma exchange	38 (28.6)
**Outcomes, *n* (%)**	
End-stage renal disease	35 (26.3)
Death	26 (19.5)

BMI, body mass index; ANCA, antineutrophil cytoplasmic antibbody; GPA, granulomatosis with polyangiitis; MPA, micropolyangeitis; BVAS, Birmingham vasculitis activity score; ICQ, interquartile; eGFR, estimated glomerular filtration rate.

**Table 2 jcm-09-03177-t002:** Description of venous thromboembolic events.

Venous Thromboembolism (VTE)	
Previous history of VTE, *n* (%)	8 (6.0)
Number of patients, *n* (%)	23 (17.3)
Number of episodes	25
Nature of VTE	
DVT leg, *n* (%)	16 (64.0)
On femoral catheter, *n* (%)	4 (25.0)
PE, *n* (%)	7 (28.0)
With DVT, *n* (%)	1 (14.3)
Others *, *n* (%)	2 (8.0)
Diagnosis of first VTE, *n* (%)	
Ultrasounds	14 (60.9)
CT scan	5 (21.7)
Chest scintigraphy	2 (8.70)
Other **	1 (4.35)
Median delay to first episode, months (ICQ)	3.03 (0.8–15)
Occurrence on initial hospital stay, *n* (%)	4 (17.4)
Occurrence within the 3 months from AAV diagnosis	11 (47.8)

AAV, ANCA-associated vasculitis; DVT, deep vein thrombosis; PE, plasma exchange; VTE, venous thromboembolism. ***** Central retinal vein thrombosis, *n* = 1; renal vein thrombosis, *n* = 1. ****** Fundus examination.

**Table 3 jcm-09-03177-t003:** Comparison of baseline characteristics of patients according to venous thromboembolic events occurrence.

	VTE		*p*
	Yes, *n* = 23	No, *n* = 110	
**Baseline characteristics**			
Gender (M/F)	11/12	72/38	0.244
Age (years)	69.7 ± 10.1	64.1 ± 14.6	0.081
BMI (kg/m^2^)	24.7 ± 3.3	24.5 ± 4.7	0.930
BMI > 30 kg/m^2^, *n* (%)	1 (4.3)	13 (11.8)	0.462
Heart disease, *n* (%)	3 (13.0)	12 (10.9)	0.724
Atrial fibrillation, *n* (%)	1 (4.3)	6 (5.4)	1.000
Past history of cancer, *n* (%)	4 (17.4)	15 (13.6)	0.743
Hypertension, *n* (%)	16 (69.6)	56 (50.9)	0.102
Diabetes mellitus, *n* (%)	5 (21.7)	13 (11.8)	0.206
History of VTE, *n* (%)	2 (8.7)	6 (20.9)	0.626
**ANCA-associated vasculitis characteristics**			
**Clinical diagnosis, *n* (%)**			
GPA/MPA, *n* (%)	8 (34.8)/15 (65.2)	37 (33.6)/73 (66.4)	0.916
Newly diagnosed, *n* (%)	22 (95.6)	100 (90.9)	0.689
**ANCA type, *n* (%)**			
PR3-ANCA	3 (13.0)	36 (32.7)	0.078
MPO-ANCA	18 (78.3)	68 (61.8)	0.134
ANCA negative	2 (8.7)	6 (5.4)	0.626
**BVAS at ANCA-GN onset**	16.7 ± 4.4	17.3 ± 6.4	0.653
**Organ involvement at diagnosis, *n* (%)**			
Cutaneous signs	4 (17.4)	21 (19.1)	1.000
Ear, nose, throat	8 (34.8)	38 (34.5)	0.983
Heart	1 (4.3)	7 (6.4)	1.000
Digestive	0 (0.0)	7 (6.4)	0.605
Lung	8 (34.8)	40 (36.4)	0.885
Neurological	2 (8.7)	17 (15.4)	0.525
Renal (at AAV diagnosis or relapse)			
Serum creatinine, µmol/L, median (ICQ)	288.0 (172–525)	260.0 (122–408)	0.421
eGFR, mL/min/1.73 m^2^, median (ICQ)	14.8 (8–39)	22.7 (11–45)	0.285
Proteinuria/creatinunuria, g/g, median (ICQ)	1.27 (0.8–2.7)	1.74 (0.9–3.1)	0.807
Proteinuria > 3 g/g, *n* (%)	3 (13.0)	28 (25.4)	0.281
Need for renal replacement therapy, *n* (%)	7 (30.4)	23 (20.9)	0.410
**Hospital stay**			
Length of stay at initial admission, days, median (ICQ)	13.0 (6–24)	12.0 (5–21)	0.564
Admission in ICU at AAV diagnosis or relapse, *n* (%)	2 (8.7)	8 (7.3)	0.823
**Biology at ANCA-GN onset**			
C-reactive protein, mg/L, median (ICQ)	56.5 (24–178)	56.9 (18–150)	0.847
Serum albumin, g/L	26.4 ± 4.8	29.8 ± 6.5	0.040
**Treatment at ANCA-GN onset, *n* (%)**			
Antiplatelet agents	2 (8.7)	29 (26.4)	0.102
Anticoagulant therapy	1 (4.3)	16 (14.5)	0.304
Statin therapy	3 (13.0)	46 (41.8)	0.009
**Induction remission regimen**			
Cyclophosphamide	20 (87.0)	96 (87.3)	1.000
Rituximab	3 (13.0)	4 (3.6)	0.099
Plasma exchange	8 (34.8)	30 (27.3)	0.468

**Table 4 jcm-09-03177-t004:** Univariate and multivariate analysis of risk factors associated with occurrence of venous thromboembolic events.

	Univariate Analysis		Multivariate Analysis	
	HR (CI)	*p*	HR (CI)	*p*
Age *	1.55 (1.06–2.26)	0.022	1.38 (0.94–2.03)	0.101
Hypertension (yes)	2.22 (0.91–5.47)	0.082	-	-
PR3-ANCA (yes)	0.40 (0.12–1.37)	0.144	-	-
Serum albumin at onset **	2.42 (1.09–5.39)	0.030	2.26 (0.98–5.23)	0.057
Antiplatelet agents (no)	3.61 (0.84–15.6)	0.085	-	-
Statin therapy (no)	5.07 (1.49–17.2)	0.009	4.73 (1.06–21.1)	0.042
Rituximab (yes)	4.52 (1.29–15.8)	0.018	2.87 (0.61–13.6)	0.182

***** Per 10 year increment. ****** Per 10 g/L decrease.

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
