# Peer review of "Incidence and Risk Factors of Venous Thromboembolic Events in Patients with ANCA-Glomerulonephritis: A Cohort Study from the Maine-Anjou Registry"

_jcm, 2020, doi:10.3390/jcm9103177_

Round 1

Reviewer 1 Report

The objective of the current study was to assess frequency and risk factors of VTE in patients with ANCA-GN.

The authors conclude that patients with ANCA-GN are at high risk for VTE, and that statin therapy might be associated with a lower risk of VTE in ANCA-GN patients

This is a well written manuscript, with some interesting observations.

I have some remarks

Regarding the registry this is qualified as "retrospective-prospective", please explain

Only 7 patients were treated with rituximab, and from table 3 it is clear that there is no difference in RTX treatment between the group with and without VTE. So I doubt wheteher this should be included in the univariate analysis. Subsequently, RTX should be left out of the concluding remark at the start of the discussion (207). Also, I doubt whether a multivariate analysis with 7 risk factors is valid in this study with only 23 events.

In the paragraph on prognosis the term "early VTE" is introduced, and it is concluded that more patients with an early VTE developed ESRD. However, this  term and analysis is not mentioned in the methods and we are not informed on the characteristics of these patients. So it should be elaborated on or left out.

Regarding the kidney biopsy histology, this is also assessed in the patients with "early VTE". This should be clarrified by the authors or also introduced in the methods. The Berden classification is not mentioned in the methods. Who scored the biopsies and what number of biopsies were representative and included in the study (what were the number of glomeruli)? In fig 3 the "mixed class" should als be included. Please explain fig3B class I to IV.

Is information available on the usage of thromboprophylaxis during admission?

Minor remark

figure 1 is filled with text and question marks...

Author Response

We would like to acknowledge Reviewers for their valuable comments that we tried to respond point by point below.

Reviewer 1

The objective of the current study was to assess frequency and risk factors of VTE in patients with ANCA-GN.

The authors conclude that patients with ANCA-GN are at high risk for VTE, and that statin therapy might be associated with a lower risk of VTE in ANCA-GN patients

This is a well written manuscript, with some interesting observations.

I have some remarks

  • Regarding the registry this is qualified as "retrospective-prospective", please explain.

We acknowledge Reviewer 1 for this comment. The registry was created in 01/01/2018 as mentioned in the Math&Meth section. Before 2018, the patients and their data were entered retrospectively in the registry. Thus, data of patients included between 2000 and 2018 were entered retrospectively in the registry. From 2018, new patients and follow-up data of patients already included in the registry before 2018 were entered prospectively.

We added one sentence in the Mat&Meth section to clarify the methodology.

Only 7 patients were treated with rituximab, and from table 3 it is clear that there is no difference in RTX treatment between the group with and without VTE. So I doubt wheteher this should be included in the univariate analysis. Subsequently, RTX should be left out of the concluding remark at the start of the discussion (207). Also, I doubt whether a multivariate analysis with 7 risk factors is valid in this study with only 23 events.

We acknowledge Reviewer 1 for these important comments. We agree with Reviewer 1 that rituximab was used in a minority of patients. However, despite the low number of patients, RTX therapy had a p-value <0.05 with VTE by univariate analysis. In consequence we added it to the multivariable analysis. It appears difficult to us to remove from the analysis RTX even if it represents a low number of patients.

Importantly, and as detailed in the Mat&Meth section, only factors with a p value <0.05 were included in the multivariable cox analysis. Thus, we included 4 variables (and not 7) in the multivariable analysis (please see table 4).

To respond to Reviewer 1 comment, we performed the multivariable cox analysis excluding RTX therapy. The table below shows the results of the multivariable analysis including age, albumin and statin therapy:

HR (CI)

P

Age (per 10 years increment)

1.39 (0.95-2.04)

0.087

Serum albumin (per 10 g/L decrease)

2.05 (0.93-4.55)

0.075

Statin therapy (no)

5.2 (1.18-22.9)

0.029

As showed in the table, the exclusion of RTX treatment does not modify significantly the results of the cox model. Thus, we preferred not to remove RTX treatment even if it represents a low number of patients.

In the paragraph on prognosis the term "early VTE" is introduced, and it is concluded that more patients with an early VTE developed ESRD. However, this  term and analysis is not mentioned in the methods and we are not informed on the characteristics of these patients. So it should be elaborated on or left out.

We totally agree with Reviewer 1 that early VTE needs to be defined in the Math&Meth section. Thus, we added the following sentence page 3, line114 to clarify: “The prognosis associated with the occurrence of VTE following the first 3 months since ANCA-GN diagnosis (early VTE) was also evaluated”

Regarding the kidney biopsy histology, this is also assessed in the patients with "early VTE". This should be clarified by the authors or also introduced in the methods. The Berden classification is not mentioned in the methods. Who scored the biopsies and what number of biopsies were representative and included in the study (what were the number of glomeruli)? In fig 3 the "mixed class" should als be included. Please explain fig3B class I to IV.

We agree with Reviewer 1 that this need to be clarified in the manuscript.

The kidney biopsies from the 4 hospital are routinely analyzed in the Pathology department of Angers University Hospital by two nephropathologists. The report is standardized and allows to determine the histopathological class according to Berden’s classification. Non representative biopsies were excluded. We choosed a cut-off of at least 7 glomeruli to consider the biopsy representative, as suggested in recent publications.

Thus, as suggested by Reviewer, we added this paragraph to the Mat&Meth section (page 3, line 105):

“Kidney biopsy results were also retrieved when available. All kidney biopsies from the four centers are analyzed centrally in the department of Pathology of the University Hospital of Angers by two nephropathologists. Only biopsies showing more than 7 glomeruli were considered in the present study. The analysis of kidney biopsies is routinely reported in a standardized pathological report allowing classification according to the Berden histopathological classification of ANCA-GN [14].”

We acknowledge Reviewer 1 for the comment according to Figure 3 that needs to be clarified. Panel A represent the % of normal glomeruli, % of glomeruli with crescents and % of fibrotic glomeruli, between patients with VTE and patients without VTE (that had kidney biopsy at ANCA-GN diagnosis). The panel B represents the % of patients with VTE according to the histological form where I is focal, II is crescentic, III is mixed, and IV is sclerotic form.

We revised the legend to improve the a better understanding of the figure.  

Is information available on the usage of thromboprophylaxis during admission?

We agree with Reviewer 1 that this an important comments, that may also represent a limitation of our study. Unfortunately, thromboprohylaxis was not collected in the registry and we feel that it will be difficult to retrieve this information on medical charts, especially for the older patients of the registry.

We underlined this limitation of our work (page 10 line 290):

“Moreover, we didn’t perform systematic screening of blood clotting and of APL antibodies and the use of thromboprophylaxis at ANCA-GN diagnosis was not collected.”

Minor remark

figure 1 is filled with text and question marks..

We added the legend to Figure. It was lacking.

Reviewer 2 Report

This study is evaluated the incidence and risk factors of venous thromboembolic events in patients with ANCA associated glomerulonephritis.
The topic is interesting, but there are several issues to consider.

1) This study is analyzed about the risk factors of venous thromboembolic events. However, common risk factors of venous thromboembolic events was not determined.
You should evaluate about known risk factors of venous thromboembolic events such as heart failure, atrial fibrillation, coronary artery disease, stroke, obesity, cancer, history of fracture (hip or leg),
the length of bed rest, and estrogen treatment.

2) serum albumin level was significantly decreased in patients with VTE although proteinuria had a lower tendency in patients with VTE.
Why do you think serum albumin is decreased in patients with VTE? Is there any difference of comorbidities or malnutrition? 

3) Please not the prevalence of nephrotic range of albuminuria.

4) You noted statin therapy was significantly higher in patients without VTE.
however, the level of LDL choleserol and triglyeride  was not noted. Is there any difference of  LDL choleserol and triglyeride levels between groups?

5) Although there was no significant difference, the use of antiplatelet agent or anticoagulant medication were higher in patients without VTE.
Please note the stratagies to start antiplatelet agent or anticoagulant medication in ANCA associated GN.

6) Rituximab treatment was associated with VTE events.
Please describe the characteristics of patients using rituximab compared than patients using other immunosuppresant.

Author Response

Reviewer 2.

This study is evaluated the incidence and risk factors of venous thromboembolic events in patients with ANCA associated glomerulonephritis.
The topic is interesting, but there are several issues to consider.

1) This study is analyzed about the risk factors of venous thromboembolic events. However, common risk factors of venous thromboembolic events was not determined.
You should evaluate about known risk factors of venous thromboembolic events such as heart failure, atrial fibrillation, coronary artery disease, stroke, obesity, cancer, history of fracture (hip or leg), 
the length of bed rest, and estrogen treatment.

We acknowledge Reviewer 2 for this important comment. As suggested by reviewer, we collected history of heart disease, of atrial fibrillation and of cancer from the registry. We added these results to table 1 and table 3. As showed in tables, VTE was not associated with these conditions. We also reported the patients with BMI > 30 kg/m2. Also, we didn’t observe any difference in VTE frequency between obese and non-obese patients. 

Unfortunately, we were not able to collect the length of bed rest, hip/leg fracture and estrogen therapy in the cohort.

2) serum albumin level was significantly decreased in patients with VTE although proteinuria had a lower tendency in patients with VTE.
Why do you think serum albumin is decreased in patients with VTE? Is there any difference of comorbidities or malnutrition? 

We acknowledge for this interesting comment. We agree with reviewer 2 that proteinuria was more pronounced in patients with VTE as compared to patients without VTE,  (even if not statistically significant). This suggests other mechanisms for hypoalbuminemia than only albuminuria. We don’t have clear explanations for that observation. However, as suggested by Reviewer 2, age and maybe malnutrition may explain the difference in albuminuria. Supporting this, VTE patients tended to be older (p=0.081). We didn’t observe any other significant differences between VTE and non VTE group that may explain hypoalbuminemia.

To underline this point, we added the following sentences to the discussion:

“In our study, proteinuria was not different between patients with and without VTE, suggesting that mechanisms other than proteinuria may be involved to explain more profound hypoalbuminemia in VTE patients.”

3) Please not the prevalence of nephrotic range of albuminuria.

We acknowledge that this is an important information to underline. Thus, we added this data to the Table 1 and Table 3, by reporting number of patients with proteinuria >3g/g

4) You noted statin therapy was significantly higher in patients without VTE.
however, the level of LDL choleserol and triglyeride  was not noted. Is there any difference of  LDL choleserol and triglyeride levels between groups?

We thanks Reviewer 2 for this comment. In fact this data was already available in supplemental Table 2. Unfortunately, these data were available only for 94 out of the 133 patients included in the study. As the reviewer can observe, there was no significant difference in cholesterol values between patients with and without statin therapy at ANCA-GN diagnosis. Thus this may suggest that patients with statins have comparable cholesterol levels as compared to patients without statins thanks to the statin therapy. Unfortunately, the triglyceride levels was not available and we could not report this information.

5) Although there was no significant difference, the use of antiplatelet agent or anticoagulant medication were higher in patients without VTE.
Please note the stratagies to start antiplatelet agent or anticoagulant medication in ANCA associated GN.

We acknowledge Reviewer 2 for this comment. In fact, we reported here the antiplatelet and anticoagulant therapy at ANCA-GN diagnosis. Thus, patients were already on these medications at ANCA-GN diagnosis. There are no specific strategies to start these agents in ANCA-GN patients. Their indication remains the same as in the general population.

6) Rituximab treatment was associated with VTE events. 
Please describe the characteristics of patients using rituximab compared than patients using other immunosuppressant.

Rituximab was used in a minority of patients of the study. This is explained by the time range of the study between 2000 and 2019. Rituximab was tested in clinical trials as induction regimen in 2010 and it started to be used in our patients in routine after 2015, mainly in patients with less severe kidney disease and in those with disease relapse.

As suggested by Reviewer 2, we compared the 7 patients that were treated with RTX as remission induction regimen to patients that were treated with other regimens (mainly cyclophosphamide).

The table below shows the main characteristics of these patients:

Rituximab

Other remission induction regimen

P

Age

70.7 ± 13.9

64.7 ± 14.1

0.278

Gender (males)

4 (57.1)

80 (63.5)

0.708

BMI

26.9 ± 3.3

24.4 ± 4.6

0.177

GFR

35.7 ± 28

33.2 ± 34

0.850

Pu g/g

1.75 ± 1.8

2.2 ± 1.9

0.565

CRP

33.4 ± 26

95.2 ± 91

0.077

BVAS

12.5 ± 3.9

17.5 ± 6.1

0.050

Follow-up length (months)

14.8 ± 5

60.2 ± 55

0.032

We acknowledge that the statistical analysis is limited by the number of patients that received RTX (n=7). However, patients treated with RTX were followed for a lower time significantly, because they were included in the last years. Moreover, they tended to have less active vasculitis as suggested by lower BVAS at ANCA-GN diagnosis.

Round 2

Reviewer 1 Report

After the revisions I am satisfied with the current form of this manuscript.